# Intrinsic Cellular Susceptibility to Barrett’s Esophagus in Adults Born with Esophageal Atresia

**DOI:** 10.3390/cancers14030513

**Published:** 2022-01-20

**Authors:** Chantal A. ten Kate, Annelies de Klein, Bianca M. de Graaf, Michail Doukas, Antti Koivusalo, Mikko P. Pakarinen, Robert van der Helm, Tom Brands, Hanneke IJsselstijn, Yolande van Bever, René M.H. Wijnen, Manon C.W. Spaander, Erwin Brosens

**Affiliations:** 1Department of Pediatric Surgery and Intensive Care Children, Erasmus MC-Sophia Children’s Hospital, 3000 CA Rotterdam, The Netherlands; c.tenkate@erasmusmc.nl (C.A.t.K.); h.ijsselstijn@erasmusmc.nl (H.I.); r.wijnen@erasmusmc.nl (R.M.H.W.); 2Department of Gastroenterology and Hepatology, Erasmus MC Cancer Institute, 3000 CA Rotterdam, The Netherlands; v.spaander@erasmusmc.nl; 3Department of Clinical Genetics, Erasmus MC Sophia Children’s Hospital, 3000 CA Rotterdam, The Netherlands; a.deklein@erasmusmc.nl (A.d.K.); b.degraaf@erasmusmc.nl (B.M.d.G.); r.vanderhelm@erasmusmc.nl (R.v.d.H.); t.brands@erasmusmc.nl (T.B.); y.vanbever@erasmusmc.nl (Y.v.B.); 4Department of Pathology, Erasmus MC, 3000 CA Rotterdam, The Netherlands; m.doukas@erasmusmc.nl; 5Department of Pediatric Surgery, University of Helsinki, Children’s Hospital, 281, 000290 Helsinki, Finland; Antti.Koivusalo@hus.fi (A.K.); Mikko.Pakarinen@hus.fi (M.P.P.)

**Keywords:** acid sensitivity, genetic predisposition, esophageal carcinoma, inflammatory response, esophagitis

## Abstract

**Simple Summary:**

We investigated the increased prevalence of Barrett’s esophagus in adults with esophageal atresia. A higher polygenic risk score and disturbances in inflammatory, stress response and oncological pathways upon acid exposure suggest a genetic susceptibility and increased induction of inflammatory processes. Although further research is required to explore this hypothesis, this could be a first-step into selecting patients that are more at risk to develop Barrett’s esophagus and/or esophageal carcinoma. Currently, an endoscopic screening and surveillance program is in practice in our institution for patients born with esophageal atresia, to early detect (pre)malignant lesions. Since recurrent endoscopies can be a burden for the patient, selecting patients by for example genetic susceptibility would allow to only include those at risk in future practice.

**Abstract:**

The prevalence of Barrett’s esophagus (BE) in adults born with esophageal atresia (EA) is four times higher than in the general population and presents at a younger age (34 vs. 60 years). This is (partly) a consequence of chronic gastroesophageal reflux. Given the overlap between genes and pathways involved in foregut and BE development, we hypothesized that EA patients have an intrinsic predisposition to develop BE. Transcriptomes of Esophageal biopsies of EA patients with BE (*n* = 19, EA/BE); EA patients without BE (*n* = 44, EA-only) and BE patients without EA (*n* = 10, BE-only) were compared by RNA expression profiling. Subsequently, we simulated a reflux episode by exposing fibroblasts of 3 EA patients and 3 controls to acidic conditions. Transcriptome responses were compared to the differential expressed transcripts in the biopsies. Predisposing single nucleotide polymorphisms, associated with BE, were slightly increased in EA/BE versus BE-only patients. RNA expression profiling and pathway enrichment analysis revealed differences in retinoic acid metabolism and downstream signaling pathways and inflammatory, stress response and oncological processes. There was a similar effect on retinoic acid signaling and immune response in EA patients upon acid exposure. These results indicate that epithelial tissue homeostasis in EA patients is more prone to acidic disturbances.

## 1. Introduction

Esophageal atresia (EA) is a congenital foregut malformation, of which improved survival rates have resulted in a growing adult population [1]. This raises new challenges in patient care as more emphasis is placed on long-term morbidities than short-term mortality. Respiratory and gastrointestinal symptoms require long-term follow-up [2]. Many adults born with EA (EA adults) suffer from chronic gastroesophageal reflux (GER), which is often underreported by patients due to an altered perception of discomfort [3]. GER can lead to reflux esophagitis, a nonspecific inflammation of the esophagus. Furthermore, the mucosal damage resulting from GER induces the replacement of esophageal squamous epithelium by gastric columnar epithelium containing goblet cells. This precursor lesion, intestinal metaplasia (IM) also known as Barrett’s esophagus (BE), can develop via dysplasia into esophageal adenocarcinoma (EAC) [4]. Basal cells at the squamous-columnar junction are the origin of the BE cell population [5]. BE tissue has crypts composed of various combinations of goblet cells, mucinous cells, endocrine cells, enterocytes and Paneth cells [6]. The prevalence of BE in EA adults is 4–5 times higher than in the general population (6.6% vs. 1.6%), and presents at a much younger median age (34 vs. 60 years) [3]. In the Erasmus MC-Sophia Children’s hospital cohort, EAC has been reported in three EA patients, and—surprisingly—also esophageal squamous cell carcinoma (ESCC) is seen more frequently in patients with EA at a younger age compared with the general population [3].

Disturbances in developmental signaling pathways are often associated with metaplasia and cancer transformation. The overlap of these pathways, disease genes and risk loci for foregut morphogenesis and BE development are suggestive of a shared etiology. During embryonic development the foregut separates into the future trachea and esophagus under the influence of spatiotemporal regulated transcriptional programs. These are regulated by gradients of morphogens that lay the blueprint for their interacting cells to develop into the various esophageal cell types and structures. Six intertwined pathways are crucial in this process: TGFB-BMP, Notch, FGF, WNT, Hedgehog and retinoic acid (RA) signaling [7]. TGFB-BMP signaling [8], SHH signaling [9] as well as RA signaling [10] are dysregulated in BE. Additionally, genome-wide association studies (GWAS) describe risk loci for the development of BE, EAC and ESCC near genes involved in these foregut developmental genes and pathways. These include *TBX5*, *GDF7*, *CRTC1*, *BARX1*, *FOXP1* and *FOXF1* [11].

Given the increased incidence of BE in EA adults, endoscopic surveillance is recommended [12]. Surveillance leads to early detection of BE or esophageal carcinoma, but could also create an unnecessary burden of repeated endoscopies for those not at risk as well as substantial added health care costs. Identifying patients at risk for developing BE could be a first step towards a tailor-made surveillance strategy. In this study, we hypothesize that patients born with EA have an increased (genetic) susceptibility for BE development. We aim to identify this predisposition by comparing risk loci burden and transcriptomes of patients with EA who have developed BE with EA patients without BE, and patients with BE without an EA history. We show that in both groups BE is histopathologically similar. However, the effect of acid reflux seems different with intrinsic cellular differences in inflammatory and stress response pathways, RA metabolism and signaling.

## 2. Materials and Methods

### 2.1. Study Population

Our institutional review board approved this case-control study (MEC-2018-1500). In our surveillance program, patients undergo upper endoscopies with histologic evaluation of biopsies taken according to a standardized protocol [3]. Biopsies and blood used in this study were retrieved from the Biobank Esophageal Atresia (MEC-2015-645) and the Biobank Barrett (MEC-2010-094). Mucosal esophageal biopsies were taken from two sites: (1) unaffected esophageal squamous cell epithelium (SQ), in EA patients taken above the original anastomosis; and (2) the GEJ or—if present—from Barrett’s mucosa. Sample extraction protocol and storage are described in Appendix A. Additionally, we genotyped six EA/BE patients from a Finnish cohort study (447/E7/2005) [13], as well as 730 ancestry matched (broadly European) unaffected controls. For the in vitro experiments we used human fibroblasts from EA patients and healthy controls. EA fibroblast lines were taken during routine diagnostic procedures. Control fibroblast lines are anonymized lines that taken previously during unrelated routine diagnostic procedures and stored for research purposes. We compared three groups of patients: patients with EA who have developed BE (EA/BE), patients with EA without BE (EA-only), and patients with BE without EA in history (BE-only) BE-only patients were matched for age and gender with EA/BE patients. See Figure 1 for study set-up.

### 2.2. Histopathological Evaluation

Hematoxylin and eosin-stained histological slides were retrieved from the archives of all patients of whom biopsies had been collected for RNA sequencing. All slides were blinded reassessed by a BE expert pathologist, according to a review-based checklist [6]. Potential differences were scored between the three groups.

### 2.3. SNP Genotyping and Calculation of Predisposing SNPs, Associated with BE

DNA extraction and quantification was done according standard procedures (see Appendix A). Processing of the SNP array genotyping chips (Infinium Global Screening Array v1.0 or v3.0 Illumina, Inc., San Diego, CA, USA) was done according to the manufacturer’s standard protocol (SM3). Output was generated using Illumina Genome studio v2.0 (Illumina, San Diego, CA, USA). Predisposition loci (and corresponding lead or proxy SNPs) associated with BE, EAC and/or ESCC were derived from literature. We used genotype data from EA/BE patients (*n* = 19), EA-only patients (*n* = 44), BE-only patients (*n* = 10) and controls (*n* = 730) to see if previously BE associated SNPs were more prevalent in EA/BE patients (see Appendix A). We used the allele counts and published ORs of the associated SNPs to calculate a polygenic risk score (PGRS) using an additive model: $\mathrm{PGRS}=\sum^{\;}\mathrm{Ln}\;\left(\mathrm{OR}\;\mathrm{risk}\;\mathrm{allele}\right)\times \;\mathrm{allele}\;\mathrm{count}$ (see Appendix A) Since we do not know if these ORs are precise enough to calculate the risk for the combination of EA and BE, we used the ORs of the associated SNPs calculated from our study population in a second calculation (see Appendix A). Using a Kruskal–Wallis test and Mann–Whitney tests, we compared the PGRS between the different groups. All statistical analyses were performed in SPSS V.25.0 (IBM, Chicago, IL, USA), with a significance level of *p* < 0.05.

### 2.4. RNA Sequencing, Differential Gene Expression and Pathway Enrichment Analysis

RNA extraction and quantification was done according standard procedures (see Appendix A). Genome-wide individual gene expression raw counts are available in Appendix A. Differential expression was calculated between (sub)groups (see Appendix A). Genes with a maximum group mean > 2, a fold change ≥ 1.5 and a false discovery rate (FDR) *p*-value < 0.05 were considered significantly differentially expressed. All differentially expressed genes per subgroup analysis were uploaded into the Ingenuity Pathway Analysis (IPA) software (Qiagen, Venlo, The Netherlands). Core analysis was performed for each (sub)group. A *p*-value of <0.05 and a Z-score of ≥2 were considered significant. Our ethics committee does not allow sharing of individual patient or control genotype information in the public domain, including sequencing reads.

### 2.5. Acid Exposure Experiments

In absence of available epithelial cells for in-vitro studies we used fibroblast. Activated fibroblasts generate extracellular matrix components and regulate inflammation [14]. There are several lines of evidence supporting a role for fibroblasts in BE proliferation and cancer [15,16]. To simulate a one-time acid reflux episode on RNA level, human fibroblasts from EA patients (*n* = 3) and healthy controls (*n* = 3) were exposed to pH adjusted cell culture medium conditions (see Appendix A). Hydrochloric acid was added to culture medium until the desired pH level was reached. Subsequently, cells were washed with phosphate buffered saline (PBS) and given standard medium. After 24 h, survival was measured (see Appendix A) with the TC20™ Automated Cell Counter (Bio-Rad Laboratories B.V., Veenendaal, The Netherlands). Cell morphology was evaluated (see Appendix A) with the Olympus IX70-S8F Inverted Fluorescence Microscope (Olympus Corporation, Tokyo, Japan). RNA was isolated and sequenced as described in Appendix A. Expression levels were compared with the RNA sequencing results of the esophageal biopsies.

### 2.6. Study Approval

The Medical Ethics Committee of the Erasmus Medical Center Rotterdam approved this study (MEC-2015-645, MEC-2010-094, MEC-2012-387). All authors had access to the study data and reviewed and approved the final manuscript.

## 3. Results

### 3.1. Study Population

Patient characteristics are depicted in Appendix A. Histopathological assessment (see Appendix A) of the biopsies is summarized in Appendix A. Columnar epithelium was present in all groups, except for two EA-only patients (see Appendix A). Since EA-only patients were selected as not having metaplasia in the distal esophagus at endoscopy, this means that most biopsies could contain part of the cardia as well. Neutrophil granulocytes were absent in the majority of EA-only patients, while a varying degree of nonspecific inflammatory cell infiltrate was present in most of them. Focusing on the characteristics of BE, IM with the presence of goblet cells was similarly present in EA/BE patients and BE-only patients. The amount of IM was larger in BE-only patients. No dysplasia was found in any of the samples.

### 3.2. SNP (Single Nucleotide Polymorphism) Genotyping

Given the limited sample size of our study population, we used ORs selected from literature to calculate the contribution of predisposing associated SNPs (polygenetic risk score, PGRS). Appendix A depicts an overview of the included SNPs and ORs. Using these ORs, we found a median PGRS of 3.24 (range 1.39–4.68) for EA/BE patients, of 2.98 (1.19–4.74) for EA-only patients and of 2.63 (1.85–3.53) for BE-only patients. There were no statistical significant differences between these groups (Figure 2A, panel a, all *p* > 0.05). When using our own data, we did find significant differences in PGRS between these groups (Figure 2A, panel b). A higher risk allele frequency was found for EA/BE patients versus BE-only patients for rs3784262 near ALDH1A2 (*p* = 0.017), and a lower risk allele frequency of rs3072 near GDF7 (*p* = 0.009) (Figure 2B and Appendix A).

### 3.3. RNA Sequencing of Esophageal Biopsy Specimens

An average of 88,378,214 reads per sample were generated (62,471,354–165,874,334). Of these reads, 98% (94.9–98.4) aligned to the human reference genome. A total of 9752 transcripts had a mean expression of ≥2 RPKM and were considered expressed. See Appendix A for the quality reports. PCA of the gene expression data confirmed clustering of the samples into the three groups (see Appendix A). PCA and quality control procedures included the exclusion of two outliers (BBE-017 and BBE-079).

### 3.4. Differential Expression and Pathway Enrichment Analysis of Esophageal Biopsy Specimens

Seven known BE disease genes [11] were differentially expressed between EA-only patients and EA/BE or BE-only patients (Figure 3 and Appendix A). Enriched pathways between EA/BE patients and BE-only patients were involved in RA signaling, stress response and inflammatory pathways, and oncological processes (see Figure 4 and Appendix A).

### 3.5. Acid Exposure Experiments

To study the effect of GER on RNA level, we simulated a reflux episode in in vitro experiments (see Figure 1). First, we optimized the acid exposure experiment (see Appendix A). Next, we exposed fibroblasts from three EA patients and three healthy controls for 30 min to medium with pH 3.5 or to normal medium (control). Cells exposed to pH 3.5 showed cell rounding and irregular cell membranes (see Appendix A). After acid exposure, there was a clear difference between upregulated and downregulated genes, both in patients and controls (see Appendix A). Ten pathways were enriched with differentially expressed genes between patients and controls (see Appendix A), that contained 244 differentially expressed genes. Subtracting the genes that were also differentially expressed without acid exposure, 81 genes of interest remained (see Appendix A). Pathway analysis of these 81 genes confirmed enrichment of pathways mostly involved in inflammatory processes (see Appendix A). Finally, we compared the results of the pathway analysis of the biopsies with those of the fibroblasts after acid exposure. Of the enriched pathways between GEJ samples of EA/BE patients and BE-only patients, 20 pathways were also enriched between fibroblasts of EA patients and controls after acid exposure (Table 1. In total, seven genes within these pathways were differentially expressed in both the GEJ samples and the acid-exposed fibroblasts (see Appendix A).

## 4. Discussion

In this first translational case-control study in adults born with esophageal atresia (EA), we compared EA patients who developed Barrett’s esophagus (BE, EA/BE) to EA patients who did not develop BE (EA-only) and BE patients without a history of EA (BE-only). Previous studies described an increased prevalence of BE in EA adults—and at a much younger age—compared with the general population [3]. Over the years, several risk loci associated with BE and/or esophageal carcinoma have been published, of which many near genes involved in foregut development [11] (S2). This overlap made us hypothesize that EA patients have an increased (genetic) susceptibility to develop BE.

### 4.1. BE Characteristics of EA/BE Patients and BE-Only Patients

There is a twenty-year difference in the age at which biopsies were taken between EA/BE patients and BE-only patients. We confirmed the lack of morphological differences between these two groups. Although endoscopic esophagitis was absent in the majority of the BE-only patients, neutrophil granulocytes were present in these patients. The typical characteristics of BE (columnar metaplasia with presence of goblet cells) were equally present, although the larger amount of IM in BE-only patients is indicative of a more advanced stage. Paneth cells were present in some patients of both groups, a variety more often reported in BE [6].

### 4.2. The Contribution of BE Associated SNPs in EA/BE Patients

The overlap of genes involved in foregut development and risk loci for BE insinuates a genetic predisposition for EA patients to develop BE. For example, *FOXF1,* which is expressed in the developing foregut [17], *BARX1,* which is expressed at the tracheoesophageal separation site and inhibits Wnt signaling [18], and *FOXP1,* which regulates esophageal muscle development [19], have all been associated with BE in previous GWAS studies [11]. *FOXP1* has also been implicated as a tumor suppressor gene in several tissues including the gastrointestinal tract [20]. The ORs of these risk loci were often small and the GWAS studies included large sets of BE patients in order to detect these predispositions.

Regardless, there seems to be an elevated risk for EA patients. EA/BE patients have a higher median PGRS compared with BE-only patients (3.24 vs. 2.63, *p* = 0.069), which was confirmed and reached significance when using ORs calculated from our study population (*p* < 0.001, see Figure 2A and Appendix A). Despite the small cohorts, the higher PGRS in EA/BE patients is suggestive for an increased predisposition, and a possible contribution for the earlier age of onset of BE in these patients. Such a relationship (higher PGRS and earlier disease onset) has been demonstrated previously in patients with atrial fibrillation [21]. However, differences in PGRS are not likely to be sufficient on their own to exclude EA patients from (pre)malignant screening protocols. Ideally, a screening algorithm would contain multiple risk factors of which the PGRS could be one. Further research would be required to confirm the impact of risk loci for BE and their potential benefit in surveillance strategies for EA patients.

Two predisposing associated SNPs proved enriched when comparing EA/BE patients with BE-only patients: rs3784262 near *ALDH1A2* (OR 3.94, *p* = 0.017) and rs3072 near *GDF7* (OR 0.22, *p* = 0.009). *ALDH1A2* (also known as *RALDH2*) is an enzyme that catalyzes the transformation of retinaldehyde into RA, a key morphogen in foregut development [22]. Lack of RA signaling results in increased TGFB-BMP signaling and hampers lung bud induction [23]. In contrast, BE is characterized by a higher expression of this enzyme, resulting in higher levels of RA [24]. *GDF7* is also a component of the TGFB-BMP signaling pathway. TGFB-BMP signaling is essential in esophageal formation by inhibiting *SOX2* in the ventral foregut [25] but also contributes to the differentiation of columnar epithelium and BE development by interacting with *CDX1* and *CDX2* [26]. Interestingly, the associated SNP *GDF7* seems a protective locus in EA/BE patients (OR 0.22, *p* = 0.009). The trends shown by these results are illustrative but more research is needed. Though EA/BE patients could have an increased genetic risk, the current sample sizes do not allow to draw firm conclusions.

### 4.3. EA/BE Patients Have Comparable Gene Expression of BE Disease Genes as BE-Only Patients

An earlier age of BE onset in EA patients could mean that epithelial homeostasis in these patients is more prone to disturbances. To investigate this, we sequenced RNA extracted from esophageal biopsies of three groups (EA/BE, EA-only and BE-only). We evaluated the expression of BE disease genes but found no difference in expression between EA/BE patients and BE-only patients. In both groups, these genes were upregulated compared to EA-only patients, indicating that the BE found in EA/BE patients is similar to the BE in BE-only patients.

### 4.4. EA/BE Patients Have an Increased Inflammatory Response

Since the expression of disease genes could not explain the earlier age of onset, we explored the complete transcriptome and corresponding differentially expressed genes and pathways. Many of the enriched pathways in EA/BE patients compared with BE-only patients, hinted at upregulated inflammatory (e.g., IL-6 signaling) and stress response pathways, downregulated oncological processes and dysregulated RA signaling (see Appendix A). Inflammatory cells produce carcinogenic compounds that can initiate DNA damage. The secretion of growth factors and cytokines increase proliferation and transition to tumor cells [27]. *SPINK1* expression itself has the potential to be a BE biomarker as it lacks expression in unaffected esophageal tissue (see Appendix A).

Human studies and in vitro experiments have shown that exposure of esophageal tissue to low pH and/or bile acids may induce cell proliferation and reduce cell apoptosis through an increased expression of cyclo-oxygenase-2 (COX-2), prostaglandin E_2_ (PGE2), mitogen-activated protein kinase (MAPK) and NF-κB pathways [28,29,30,31]. In our data, p38 MAPK Signaling and NF-κB Signaling are upregulated in EA/BE patients compared with BE-only patients. Given their proliferative and anti-apoptotic role, these pathways could be valuable for BE staging. Quante and coworkers showed that transgenic mice, overexpressing human *IL-1**β*, presented with chronic inflammation, BE and esophageal dysplasia. Oral exposure to bile acids led to elevated IL-6 levels, accelerating BE development and progression into EAC, and implicating an IL-1β-IL-6 signalling cascade [32]. Clinical management of BE is focused around chemical inhibition of acid exposure and decrease of inflammation. Inhibition of gastric acid secretion with proton pump inhibitors (PPIs) reduces the transition to dysplasia in BE patients [33] and a combination of non-steroidal anti-inflammatory drugs (NSAIDs) and statins may reduce neoplastic progression [34]. Recently, it has been shown that the combination of high-doses esomeprazole and aspirin reduces high-grade dysplasia and EAC in BE patients [35]. Given the potentially altered response to acid in EA patients, the effectiveness of PPIs and NSAIDs in this population warrant further investigation.

Furthermore, stress response pathways are upregulated. Cholecystokinin/Gastrin-mediated Signaling is an activator of actin stress fiber formation and intertwined with stress response pathways as p38 MAPK Signaling, Sphingosine-1-phosphate Signaling and Signaling by Rho Family GTPases. These processes may lead to the conversion of squamous epithelium to columnar metaplasia. Another study showed that low pH and/or bile acids can induce oxidative stress, which causes DNA damage [36]. In combination with reduced apoptosis this can lead to dysplasia. When this is followed by neoplastic progression BE can develop into EAC.

### 4.5. Dysregulation of RA Metabolism and Signaling

RA is increased in BE and works—like bile acids—through the RXR receptors to transform squamous epithelium to columnar epithelium [10]. LXR/RXR activation, involved in RA mediated gene activation, is downregulated in EA/BE patients compared with BE-only patients. Retinol biosynthesis is also downregulated, whilst its downstream processes in all trans RA synthesis (Retinoate Biosynthesis I) are upregulated. Peroxisome proliferator-activated receptors (PPARs) are transcription factors activated by RA, generally upregulated in BE [37], but downregulated in EA/BE patients. Like discussed above, the downregulation of these pathways could indicate that BE-only patients are at a more advanced stage than EA/BE patients. Given the clinical differences (age and length of BE) between these patients, this does make sense.

### 4.6. Downregulation of the Hippo/YAP Pathway

Downregulation of oncological pathways in EA/BE patients could be indicative of either a decreased progression rate to dysplasia or a less advanced state of progression compared with BE-only patients. The Hippo/YAP pathway is important in cell proliferation, survival, and differentiation. Yes-association protein (YAP) expression is associated with dysplasia and adenocarcinoma [38]. Hippo signaling is involved in cell contact inhibition [39] as is Aryl Hydrocarbon Receptor Signaling [40]. Hippo activation (and YAP inactivation) is necessary for programmed cell death after detachment from the extracellular matrix [41]. Therefore, downregulation of this pathway could (in theory) decrease anoikis and increase the risk of tumor cell metastasis.

### 4.7. EA Patients Seem to Be More Sensitive to Acid Reflux Exposure

EA patients are earlier in life and more frequently exposed to GER. Chronic GER could be a consequence of the surgical repair: the lower esophageal sphincter is often retracted above the diaphragm, resulting in the loss of the natural reflux barrier function of the GEJ [42]. Other factors contributing to GER are impaired motility, delayed bolus clearance and delayed gastric emptying [43]. There seems to be a direct relationship of these symptoms with EA, as Adriamycin induced EA rats have impaired esophageal relaxation and a decreased number of ganglia and nerve fibers in the esophageal myenteric plexus [44]. The prevalence of mucosal damage is related to the level of pH exposure and to the composition of the acid reflux [45]. Animal studies have shown that acid fluids can activate pepsin, which inflicts injury and leads to mucosal damage [46].

We speculated that GER could result in an upregulation of inflammatory pathways. Additionally, EA patients could have a predisposition that makes them more sensitive to acid reflux than the general population. To explore these hypotheses, we performed in vitro experiments to simulate a one-time reflux episode in fibroblasts of EA patients and healthy controls. The enriched pathways of the GEJ biopsies of EA/BE patients showed an overlap with the enriched pathways of the fibroblasts of EA patients after acid exposure—but not with those of healthy controls. These overlapping pathways were again mostly involved inflammatory or oncological processes. For example, LXR/RXR Activation, PPAR Signaling and Retinol Biosynthesis were also enriched in fibroblasts of EA patients after acid exposure, hinting at intrinsic disturbances of RA signaling in EA patients under the influence of GER.

We do not know of the three patients used in the in vitro experiment will develop BE in time as the fibroblasts are derived of patients currently aged 29, 30 and 39 years old. It is, however, interesting that we could detect a similar predisposition in just 3 EA patients, and as a general response (in fibroblasts) to acid.

### 4.8. Strengths and Limitations

The main strength of this study is the broad investigative approach by combining histology, genotype, transcriptome and in vitro results. Some limitations should be addressed. First, due to the relative low incidence of EA and corresponding small sample sizes, we mostly observed trends and more EA/BE patients are needed to draw more robust conclusions. At this point, the difference in gene expression between EA/BE patients and EA-only patients is negligible. This could be due to the fact that most biopsies could contain part of the cardia. However, the power would increase substantially if we would know which EA patients have not developed BE throughout their life, as the current EA-only population is a mixture of patients who have not yet and will never develop BE. Second, EA is a heterogeneous disease. Our study population included both patients with isolated EA and patients with syndromes or multiple anomalies. This phenotypic heterogeneity might also be the results of a genetic heterogeneity. Thirdly, BE can present as a heterogeneous metaplastic mosaic, consisting of multiple individual crypts that arose from independent clones [47], which have distinct ploidies, copy number variations (CNV) and point mutations [48]. Heterogeneity in these crypts pose a risk of sampling error. Even within long segment BE, IM can be focally distributed [49]. Recent progress in genetic analysis of BE stem cells and EAC indicates that there are patient-specific driver genes affected in both the precursor lesion [50] and subsequent cancer of the esophagus [51]. Perhaps the heterogeneous background of de novo mutations [52] and de novo CNVs [53] in EA contributes to this patient-centred susceptibility. This could have created larger variances in gene expression per evaluated group. Subsequent experiments using single-cell sequencing of definite IM could reveal differences between patients that cannot be detected in whole biopsy specimens. Lastly, morphological differences were absent. However, segment length differences could be related to a difference in disease stage [54] and impact gene networks are prone to disturbances.

## 5. Conclusions

Altered regulation of p38 MAPK, NF-κB and RA signaling could have implications for (or be related to) the dysplastic progression. If Hippo/YAP signaling remains downregulated upon progression to cancer, the metastasis risk could be higher in EA patients due to reduced anoikis. An increased PGRS and upregulation of inflammatory pathways hint at a multifactorial contribution underlying the earlier age of onset of BE in EA patients. We did not evaluate mechanical factors such as loss of the natural reflux barrier due to the surgical repair and clinical factors such as impaired esophageal motility. These factors increase the level of acid exposure and likely add to the effect of risk loci and primed inflammatory pathways.

## Figures and Tables

**Figure 1 cancers-14-00513-f001:**
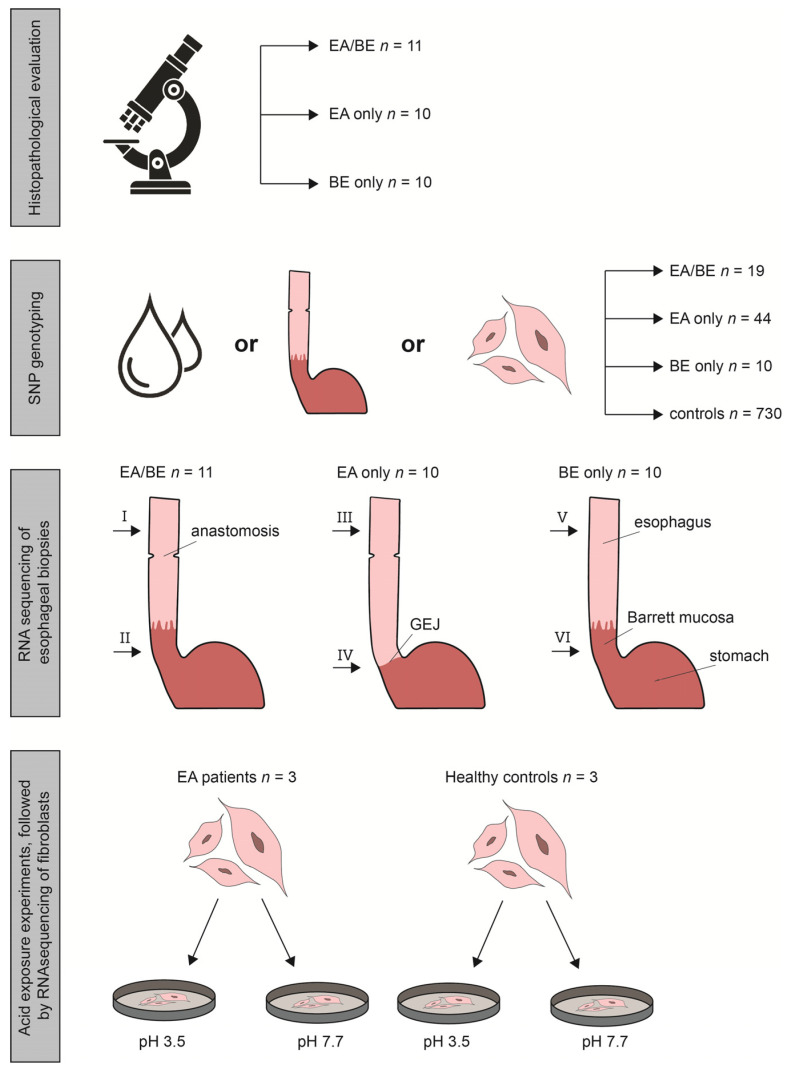
Schematic overview of the study set-up and number of patients included in each part. We compared three groups of patients: patients with esophageal atresia (EA) who have developed Barrett’s esophagus (BE, EA/BE), patients with EA without BE (EA-only), and patients with BE without EA in history (BE-only). BE-only patients were matched for age and gender with EA/BE patients. Roman numerals I to VI indicate the subgroups, based on the location of the biopsies. GEJ = gastroesophageal junction.

**Figure 2 cancers-14-00513-f002:**
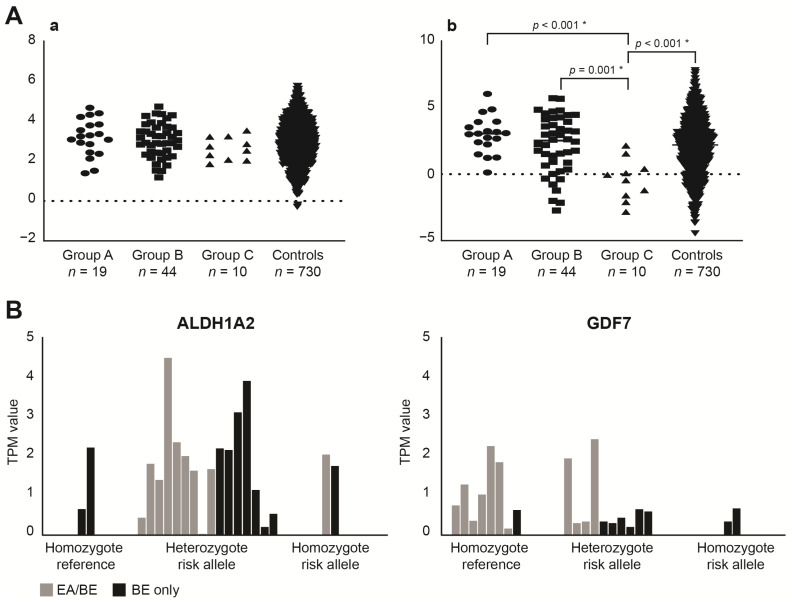
(**A**) Polygenic risk scores (PGRS) per patient. Group A = patients with esophageal atresia (EA) and Barrett’s esophagus (BE), group B = patients with EA without BE, group C = patients with BE without EA in history. Panel a (left) are PGRS based on odds ratios (ORs) selected from the literature. No statistical significant differences between the groups were observed. Panel b (right): PGRS based on ORs calculated from our study population. We found a median PGRS of 3.05 (range 0.14–6.04) for EA/BE patients, of 2.52 (−2.73–5.72) for EA-only patients and of −0.24 (−2.83–2.15) for BE-only patients. A Kruskal–Wallis test revealed a significant difference in PGRS based on ORs calculated from our study population between the four groups (*p* = 0.001). T-statistics indicated a difference between BE-only patients versus EA/BE patients (*p* < 0.001), EA-only patients (*p* = 0.001) and controls (*p* < 0.001). Asterisk (*) indicates significance *p* < 0.05. (**B**) Gene expression levels for ALDH1A2 and GDF7 per patient, sorted based on the genotype of the patients. A higher risk allele frequency was found for EA/BE patients versus BE-only patients for rs3784262 near ALDH1A2 (*p* = 0.017) and a putative protective allele for rs3072 near GDF7 (*p* = 0.009). Looking at gene expression levels, GDF7 has slightly elevated TPM values for patients homozygote for the reference allele. No significant differences could be detected for these two associated SNPs. TPM = transcripts per million, EA = esophageal atresia, BE = Barrett’s esophagus. Complete results can be found in Appendix A.

**Figure 3 cancers-14-00513-f003:**
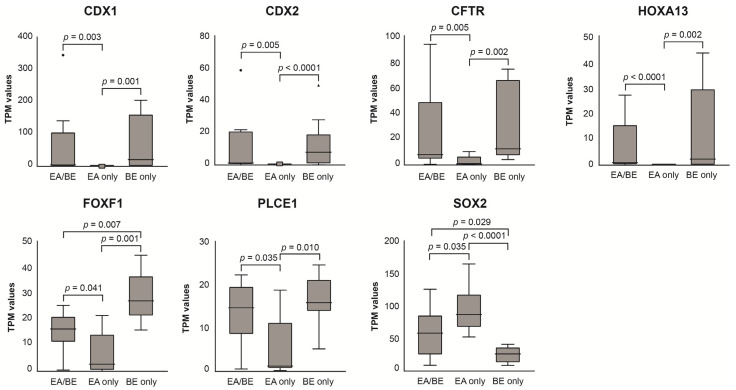
Gene expression levels per group for selected disease genes, involved in foregut morphogenesis and/or associated with Barrett’s esophagus in literature, presented as median (interquartile range) with minimum and maximum values. We compared biopsies of the gastroesophageal junction between three groups of patients: patients with esophageal atresia (EA) who have developed Barrett’s esophagus (BE) (EA/BE, *n* = 11), patients with EA without BE (EA-only, *n* = 10), and patients with BE without EA in history (BE-only, *n* = 10). TPM = transcripts per million, EA = esophageal atresia, BE = Barrett’s esophagus.

**Figure 4 cancers-14-00513-f004:**
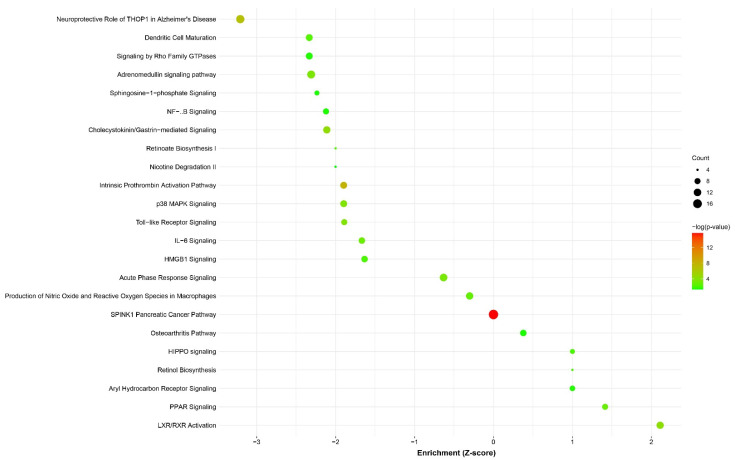
Bubble plot of canonical pathways, significantly enriched by differentially expressed genes, between gastroesophageal junction (GEJ) samples of group A (esophageal atresia (EA) with Barrett’s esophagus (BE)) and GEJ samples of group C (BE-only). The color and size of the dots represent the range of the *p*-value and the number of molecultes mapped to the indicated pathways. Settings: *p*-value < 0.05 (=−log(*p*-value) > 1.3), z-score < −2 or >2. SPINK1 Pancreatic Cancer Pathway is also the only significantly upregulated pathway, when comparing group A (EA/BE) with group C (BE-only). Plotted by http://www.bioinformatics.com.cn (accessed on 24 November 2021), a free online platform for data analysis and visualization.

**Table 1 cancers-14-00513-t001:** Overlap between canonical pathways, significantly enriched by differentially expressed genes, in the esophageal biopsy specimens versus the acid-exposed and non-exposed fibroblasts. II = gastroesophageal junction (GEJ) samples from group A (esophageal atresia (EA) and Barrett’s esophagus (BE)), VI = GEJ samples from group C (BE-only). *n* = total number of canonical pathways significantly enriched by differentially expressed genes. N/A = not applicable, z-score could not be calculated. Grey box indicates that a pathway was not present in the results of that pathway analysis.

	Esophageal Biopsy Specimens		Fibroblasts from Acid Exposure experiment					
	II vs. VI (*n* = 353)		EA Patients vs. Controls (Acid-Exposed) (*n* = 258)		EA Patients vs. Controls (Non-Exposed) (*n* = 314)		Acid-Exposed vs. Non-Exposed (All Samples) (*n* = 578)	
Canonical Pathways	*−log(p-Value)*	*Z-Score*	*−log(p-Value)*	*Z-Score*	*−log(p-Value)*	*Z-Score*	*−log(p-Value)*	*Z-Score*
Agranulocyte Adhesion and Diapedesis	1.69	N/A	-	-	1.52	N/A	-	-
Altered T Cell and B Cell Signaling in Rheumatoid Arthritis	3.22	N/A	2.57	N/A	2.05	N/A	-	-
Atherosclerosis Signaling	4.93	N/A	2.04	N/A	2.23	N/A	-	-
Cholecystokinin/Gastrin-mediated Signaling	4.38	2.111	2.35	0	1.39	N/A	-	-
Communication between Innate and Adaptive Immune Cells	2.39	N/A	2.47	N/A	-	-	-	-
Dendritic Cell Maturation	2.27	2.333	4.600	−0.707	2.19	−1.633	-	-
Extrinsic Prothrombin Activation Pathway	1.36	N/A	2.34	N/A	-	-	-	-
Glucocorticoid Receptor Signaling	4.51	N/A	1.53	N/A	2.03	N/A	-	-
Graft-versus-Host Disease Signaling	4.23	N/A	3.600	N/A	-	-	-	-
HMGB1 Signaling	2.09	1.633	2.37	N/A	-	-	-	-
IL-6 Signaling	2.89	1.667	1.33	N/A	-	-	5.02	2.117
Intrinsic Prothrombin Activation Pathway	7.92	1.897	2.61	N/A	-	-	-	-
LXR/RXR Activation	4.31	−2.111	2.12	−1	-	-	-	-
MSP-RON Signaling Pathway	4.58	N/A	1.44	N/A	1.81	N/A	-	-
Osteoarthritis Pathway	1.44	−0.378	-	-	6.98	−1.265	-	-
PPAR Signaling	2.81	−1.414	2.33	0	-	-	2.57	−2.524
Production of Nitric Oxide and Reactive Oxygen Species in Macrophages	2.68	0.302	1.48	N/A	-	-	-	-
Retinol Biosynthesis	1.95	−1	1.49	N/A	-	-	-	-
Role of Macrophages, Fibroblasts and Endothelial Cells in Rheumatoid Arthritis	2.59	N/A	3.710	N/A	3.87	N/A	-	-
Role of Osteoblasts, Osteoclasts and Chondrocytes in Rheumatoid Arthritis	3.10	N/A	3.280	N/A	4.53	N/A	-	-
Sphingosine-1-phosphate Signaling	1.44	−2.236	1.40	N/A	2.38	−1	-	-

## Data Availability

All transcriptome count data is in the Appendix A. Our ethics committee does not allow sharing of individual patient or control genotype information in the public domain.

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
