# Peer review of "Intrinsic Cellular Susceptibility to Barrett’s Esophagus in Adults Born with Esophageal Atresia"

_cancers, 2022, doi:10.3390/cancers14030513_

Round 1

Reviewer 1 Report

In this manuscript ten Kate et al. hypothesized that patients with esophageal atresia (EA) have an increased genetic susceptibility for the development of Barrett’s esophagus (BE) and esophageal adenocarcinoma (EAC). Following this hypothesis, the authors carried out three main experiments: (i) They tested whether genetic risk alleles of variants that have been previously shown to be BE/EAC-associated in a genome-wide association study (GWAS) are enriched on the single and multi-marker level in EA/BE cases (n=19) compared to EA only (n=44) and BE only cases (n=10) as well as controls (n=730). (ii) They analyzed whether there are any expression differences in esophageal and/or Barrett’s tissue between EA/BE cases (n=11), EA only cases (n=10) and BE only cases (n=10). (iii) Finally, they exposed fibroblasts from three EA cases and three controls to medium with pH 3.5 and to normal medium (control) in order to see transcriptome differences between both groups.

The hypothesis the authors raised is interesting. However, the manuscript is difficult to follow in several sections and the conclusions are not clear in some parts.

  • The polygenic risk scores (PRS) derived from the BE/EAC GWAS do not differ between EA/BE cases, EA only cases, BE only cases and controls (Fig 2A Panel a). However, in Fig 2A Panel b, which is not mentioned in the text of the manuscript, the authors see PRS differences between groups. Here, the PRS is based on odds ratios (OR) calculated from the cohorts that are part of the study. It is not clear what the authors have done, but it seems that they performed association analyses using their small cohorts and used OR that are not referring to the BE/EAC GWAS risk alleles. The authors should specify and justify the methods they used in Fig 2A Panel b.
  • In Fig 2B the authors claim that they found higher risk allele frequencies for the BE/EAC risk variants near ALDH1A2 and GDF7 among patients with EA/BE versus BE only patients (lines 178-180). That is misleading as the risk allele frequency for the risk variant near GDF7 is more common among patients with BE only.
  • Taking both of the above-mentioned points into account, it does not seem correct to claim that EA/BE patients are at greater genetic BE/EAC risk (subheading in line 256).
  • In line 217 the authors are referring to Fig 3. This most probably refers to Fig 1.
  • In line 295 the authors are referring to Fig 2. This most probably refers to Fig 3.
  • The authors should use the term “variants” instead of “haplotypes” in paragraph 4.2. and elsewhere in the manuscript. In addition, the authors should clarify that not genes show association to BE/EAC in the GWAS, but rather genetic variants which are near to the genes.

Author Response

In this manuscript ten Kate et al. hypothesized that patients with esophageal atresia (EA) have an increased genetic susceptibility for the development of Barrett’s esophagus (BE) and esophageal adenocarcinoma (EAC). Following this hypothesis, the authors carried out three main experiments: (i) They tested whether genetic risk alleles of variants that have been previously shown to be BE/EAC-associated in a genome-wide association study (GWAS) are enriched on the single and multi-marker level in EA/BE cases (n=19) compared to EA only (n=44) and BE only cases (n=10) as well as controls (n=730). (ii) They analyzed whether there are any expression differences in esophageal and/or Barrett’s tissue between EA/BE cases (n=11), EA only cases (n=10) and BE only cases (n=10). (iii) Finally, they exposed fibroblasts from three EA cases and three controls to medium with pH 3.5 and to normal medium (control) in order to see transcriptome differences between both groups.

The hypothesis the authors raised is interesting. However, the manuscript is difficult to follow in several sections and the conclusions are not clear in some parts.

The polygenic risk scores (PRS) derived from the BE/EAC GWAS do not differ between EA/BE cases, EA only cases, BE only cases and controls (Fig 2A Panel a). However, in Fig 2A Panel b, which is not mentioned in the text of the manuscript, the authors see PRS differences between groups. Here, the PRS is based on odds ratios (OR) calculated from the cohorts that are part of the study. It is not clear what the authors have done, but it seems that they performed association analyses using their small cohorts and used OR that are not referring to the BE/EAC GWAS risk alleles. The authors should specify and justify the methods they used in Fig 2A Panel b.

Thank you for the careful appraisal of our manuscript. Panel a and panel b of Figure 2A do indeed show two different methods. For panel A, we calculated the polygenic risk scores (PGRS) based on odds ratios (ORs) published in GWAS studies on Barrett’s esophagus. However, we do not know if these ORs are precise enough to calculate the risk for the combination of esophageal atresia (EA) and Barrett’s esophagus. Therefore, we made a second calculation in panel b, for which we calculated the PGRS based on ORs calculated from our own study population, as we describe in the footnote of Figure 2. This calculation does show significant differences but is underpowered.

Our study population is indeed too small to make firm conclusions. But these results are more to illustrate that more research is needed as they do seem to show a trend when using our own data. As EA is a relatively rare disorder – and the combination of EA with Barret’s esophagus is even more rare – it is difficult to find sufficient numbers of patients for these cohorts. We therefore do not make firm conclusions and suggest further research.

In retrospect, we agree that this might not be perfectly clear in the manuscript. To clarify this, we have specified our methods in line 132-134: “Since we do not know if these ORs are precise enough to calculate the risk for the combination of EA and BE, we used the ORs of the associated SNPs calculated from our study population in a second calculation.” We have explained the result of Figure 2A, panel b in the text in line 186-187: “When using our own data, we did find significant differences in PGRS between these groups (Figure 2A, panel b).” Last, we added the interpretation of these results – in light of the small cohorts – in 286: “Despite the small cohorts, …”

In Fig 2B the authors claim that they found higher risk allele frequencies for the BE/EAC risk variants near ALDH1A2 and GDF7 among patients with EA/BE versus BE only patients (lines 178-180). That is misleading as the risk allele frequency for the risk variant near GDF7 is more common among patients with BE only.

We changed the text to a lower risk allele frequency in line 187-189: “A higher risk allele frequency was found for EA/BE patients versus BE only patients for rs3784262 near ALDH1A2 (p=0.017), and a lower risk allele frequency of rs3072 near GDF7 (p=0.009) (Figure 2B).”

Taking both of the above-mentioned points into account, it does not seem correct to claim that EA/BE patients are at greater genetic BE/EAC risk (subheading in line 256).

We have adjusted the subheading in line 273-274: “The contribution of BE associated SNPs in EA/BE patients” Moreover, we have added the following sentences to line 306-308: “The trends shown by these results are illustrative but more research is needed. Though EA/BE patients could have an increased genetic risk, the current sample sizes do not allow to draw firm conclusions.”

In line 217 the authors are referring to Fig 3. This most probably refers to Fig 1.

Thank you for pointing this out. We had adjusted the reference to Figure 1, as it should have been.

In line 295 the authors are referring to Fig 2. This most probably refers to Fig 3.

Thank you for pointing this out. We had adjusted the reference to Figure 3, as it should have been.

The authors should use the term “variants” instead of “haplotypes” in paragraph 4.2. and elsewhere in the manuscript. In addition, the authors should clarify that not genes show association to BE/EAC in the GWAS, but rather genetic variants which are near to the genes.

We partly agree with the reviewer. Most of the SNPs discovered in GWAS studies are not the causal variant but are associated to a haplotype that holds a causal variant. In this study, we used the lead SNPs for the analysis. For most of these SNPs there is no evidence that these themselves are causal variants. Therefore, we changed ‘haplotypes’ to ‘associated SNPs’ where applicable throughout the manuscript.

Reviewer 2 Report

Chantal A. ten Kate et al. wrote a research article of intrinsic cellular susceptibility to Barrett’s esophagus in adults born with esophageal atresia, which indicates that epithelial tissue homeostasis in esophageal atresia patients is more prone to acidic disturbances.

While the article is well written and contains new and interesting information, I think that this report should be more attractive for the readers if further information is added as indicated below.

My minor comment is as follows.

  1. In the Supplementary Table S1.3., the magnification of the picture of intestinal metaplasia 70% is different from other pictures and the intestinal metaplasia presentation remains ambiguous. The authors should present more typical picture with the same magnification or add the scale bars in every pictures. The authors should also write the total magnification of the picture in the materials and methods section.

  1. In the Supplementary Figure S7.1., is the magnification ×20 true? Please confirm. The total magnification should be written.

  1. The authors should clearly explain how they obtained the healthy control human fibroblasts which were used in the acid exposure experiments. Is that also approved by Medical Ethics Committee with written informed consent? Please check.

Author Response

Chantal A. ten Kate et al. wrote a research article of intrinsic cellular susceptibility to Barrett’s esophagus in adults born with esophageal atresia, which indicates that epithelial tissue homeostasis in esophageal atresia patients is more prone to acidic disturbances.

While the article is well written and contains new and interesting information, I think that this report should be more attractive for the readers if further information is added as indicated below.

My minor comment is as follows.

  1. In the Supplementary Table S1.3., the magnification of the picture of intestinal metaplasia 70% is different from other pictures and the intestinal metaplasia presentation remains ambiguous. The authors should present more typical picture with the same magnification or add the scale bars in every pictures. The authors should also write the total magnification of the picture in the materials and methods section.

Thank you for your positive response. We are happy to hear you find the information in this manuscript of interest. We have changed the illustrative images of intestinal metaplasia to more illustrative slides, and have added the magnification to each image. The histopathological evaluation is dynamic process, in which the pathologist reviews all available slides of a biopsy in different magnifications. Therefore, it is not possible to add one total magnification to the Methods section.

2.In the Supplementary Figure S7.1., is the magnification ×20 true? Please confirm. The total magnification should be written.

We confirm that the 20x magnification is correct.

  1. The authors should clearly explain how they obtained the healthy control human fibroblasts which were used in the acid exposure experiments. Is that also approved by Medical Ethics Committee with written informed consent? Please check.

The fibroblasts were obtained during routine diagnostic procedures, which the Medical Ethics Committee approved in 2012 (MEC-2012-387). The opt-out procedure was applied. We have clarified this in line 102-105: “EA fibroblast lines were taken during routine diagnostic procedures. Control fibroblast lines are anonymized lines that taken previously during unrelated routine diagnostic procedures and stored for research purposes.”

Reviewer 3 Report

1.In the box chart in figure3, it is best to give the P value for pairwise comparison and explain each part of the box chart.

2.When displaying the results of pathway enrichment, the enrichment bubble chart may be a better form of expression than the table.

3. Why only fibroblasts but not epithelial cells were exposed to median with different pH levels? 

4. Table S1.3 should be a figure but not a table. 

5. As shown in Table 2, some pathways involving immune-response, such as dendritic cell maturation, altered T and B cell signaling, production of NO and ROS in macrophages were enriched in fibroblast after acid exposure. Please explain why immune cells-related pathways would happen in fibroblasts. 

Author Response

  1. In the box chart in figure3, it is best to give the P value for pairwise comparison and explain each part of the box chart.

Thank you for reading our manuscript. We have added the p-values to Figure 3 for the significant comparisons, and explained the boxplot further by adding additional information to the footnote.

  1. When displaying the results of pathway enrichment, the enrichment bubble chart may be a better form of expression than the table.

We have replaced Table 1 with a pathway enrichment bubble chart (Figure 4).

  1. Why only fibroblasts but not epithelial cells were exposed to median with different pH levels?

Thank you for this interesting question, which has crossed our minds as well. Unfortunately, epithelial cells were not available for culturing. We used esophageal biopsies for RNA sequencing from two biobanks, but this material was snap-frozen and therefore not viable for cell culture. Fibroblast cell lines were available from routine diagnostic procedures (line 102-105). Since we believe it would not be ethical to have patient undergo a gastroscopy just to retrieve research material, we decided to use fibroblasts instead of epithelial cells for the acid exposure experiments. There is evidence that fibroblast generate extracellular matrix components, regulate inflammation and play a role in Barrett’s esophagus proliferation and cancer. We have added these arguments – including corresponding literature – for our decision in line 150-153: “In absence of available epithelial cells for in-vitro studies we used fibroblast. Activated fibroblasts generate extracellular matrix components and regulate inflammation14. There are several lines of evidence supporting a role for fibroblasts in BE proliferation and cancer15, 16.” There is a lot of interplay between the immune system and Cancer Associated Fibroblasts (Sahai et al., Nature Reviews Cancer, 2020).

  1. Table S1.3 should be a figure but not a table.

We have changed Table S1.3 to a figure.

  1. As shown in Table 2, some pathways involving immune-response, such as dendritic cell maturation, altered T and B cell signaling, production of NO and ROS in macrophages were enriched in fibroblast after acid exposure. Please explain why immune cells-related pathways would happen in fibroblasts.

We have added the following sentence to line 150-153 of the Methods section: “Activated fibroblasts generate extracellular matrix components and regulate inflammation14. There are several lines of evidence supporting a role for fibroblasts in BE proliferation and cancer15, 16.” There is increasing evidence that fibroblasts are very versatile cells and support a lot of functions in other cells. There is a lot of interplay between the immune system and Cancer Associated Fibroblasts (Sahai et al., Nature Reviews Cancer, 2020).

Our hypothesis was that, whilst not immune cells, specific genes – and as a consequence pathways in a pathway analysis – would be stimulated by acid exposure in the fibroblasts, and EA cells would be more prone to this effect. This was done as a confirmation experiment mimicking the effect of acid exposure seen in gastro-esophageal reflux disease in EA/BE patients. There, the exposure is  to the mix of cells in biopsies. For instance, NOS and ROS signaling pathways are present in fibroblasts and these pathways have much overlap with NOS and ROS signaling in macrophages. In the pathway analysis “production of NO and ROS in macrophages” would pop up in fibroblasts, because the same genes are stimulated, not because of the same cell are activated.

Round 2

Reviewer 1 Report

In this manuscript ten Kate et al. hypothesized that patients with esophageal atresia (EA) have an increased genetic susceptibility for the development of Barrett’s esophagus (BE) and esophageal adenocarcinoma (EAC). Following this hypothesis, the authors carried out three main experiments: (i) They tested whether genetic risk alleles of variants that have been previously shown to be BE/EAC-associated in a genome-wide association study (GWAS) are enriched on the single and multi-marker level in EA/BE cases (n=19) compared to EA only (n=44) and BE only cases (n=10) as well as controls (n=730). (ii) They analyzed whether there are any expression differences in esophageal and/or Barrett’s tissue between EA/BE cases (n=11), EA only cases (n=10) and BE only cases (n=10). (iii) Finally, they exposed fibroblasts from three EA cases and three controls to medium with pH 3.5 and to normal medium (control) in order to see transcriptome differences between both groups.

The hypothesis the authors raised is interesting. However, the manuscript is difficult to follow in several sections and the conclusions are not clear in some parts.

  • The polygenic risk scores (PRS) derived from the BE/EAC GWAS do not differ between EA/BE cases, EA only cases, BE only cases and controls (Fig 2A Panel a). However, in Fig 2A Panel b, which is not mentioned in the text of the manuscript, the authors see PRS differences between groups. Here, the PRS is based on odds ratios (OR) calculated from the cohorts that are part of the study. It is not clear what the authors have done, but it seems that they performed association analyses using their small cohorts and used OR that are not referring to the BE/EAC GWAS risk alleles. The authors should specify and justify the methods they used in Fig 2A Panel b.
  • In Fig 2B the authors claim that they found higher risk allele frequencies for the BE/EAC risk variants near ALDH1A2 and GDF7 among patients with EA/BE versus BE only patients (lines 178-180). That is misleading as the risk allele frequency for the risk variant near GDF7 is more common among patients with BE only.
  • Taking both of the above-mentioned points into account, it does not seem correct to claim that EA/BE patients are at greater genetic BE/EAC risk (subheading in line 256).
  • In line 217 the authors are referring to Fig 3. This most probably refers to Fig 1.
  • In line 295 the authors are referring to Fig 2. This most probably refers to Fig 3.
  • The authors should use the term “variants” instead of “haplotypes” in paragraph 4.2. and elsewhere in the manuscript. In addition, the authors should clarify that not genes show association to BE/EAC in the GWAS, but rather genetic variants which are near to the genes.

Reviewer 3 Report

The authors have replied to all the comments.